# Design, Synthesis, Docking, DFT, MD Simulation Studies of a New Nicotinamide-Based Derivative: In Vitro Anticancer and VEGFR-2 Inhibitory Effects

**DOI:** 10.3390/molecules27144606

**Published:** 2022-07-19

**Authors:** Eslam B. Elkaeed, Reda G. Yousef, Hazem Elkady, Ibraheem M. M. Gobaara, Bshra A. Alsfouk, Dalal Z. Husein, Ibrahim M. Ibrahim, Ahmed M. Metwaly, Ibrahim H. Eissa

**Affiliations:** 1Department of Pharmaceutical Sciences, College of Pharmacy, AlMaarefa University, Riyadh 13713, Saudi Arabia; ikaeed@mcst.edu.sa; 2Pharmaceutical Medicinal Chemistry & Drug Design Department, Faculty of Pharmacy (Boys), Al-Azhar University, Cairo 11884, Egypt; redayousof@azhar.edu.eg (R.G.Y.); hazemelkady@azhar.edu.eg (H.E.); 3Zoology Department, Faculty of Science (Boys), Al-Azhar University, Cairo 11884, Egypt; ibraheemgobaara@azhar.edu.eg; 4Department of Pharmaceutical Sciences, College of Pharmacy, Princess Nourah bint Abdulrahman University, P.O. Box 84428, Riyadh 11671, Saudi Arabia; baalsfouk@pnu.edu.sa; 5Chemistry Department, Faculty of Science, New Valley University, El-Kharja 72511, Egypt; dalal_husein@sci.nvu.edu.eg; 6Biophysics Department, Faculty of Science, Cairo University, Cairo 12613, Egypt; iabdelmagid@sci.cu.edu.eg; 7Pharmacognosy and Medicinal Plants Department, Faculty of Pharmacy (Boys), Al-Azhar University, Cairo 11884, Egypt; 8Biopharmaceutical Products Research Department, Genetic Engineering and Biotechnology Research Institute, City of Scientific Research and Technological Applications (SRTA-City), Alexandria 21934, Egypt

**Keywords:** nicotinamide, VEGFR-2 inhibitors, molecular docking, MD simulations, MM-GBSA, PLIP, DFT, ADMET, in vitro antiproliferative

## Abstract

A nicotinamide-based derivative was designed as an antiproliferative VEGFR-2 inhibitor with the key pharmacophoric features needed to interact with the VEGFR-2 catalytic pocket. The ability of the designed congener ((*E*)-N-(4-(1-(2-(4-benzamidobenzoyl)hydrazono)ethyl)phenyl)nicotinamide), compound **10**, to bind with the VEGFR-2 enzyme was demonstrated by molecular docking studies. Furthermore, six various MD simulations studies established the excellent binding of compound **10** with VEGFR-2 over 100 ns, exhibiting optimum dynamics. MM-GBSA confirmed the proper binding with a total exact binding energy of −38.36 Kcal/Mol. MM-GBSA studies also revealed the crucial amino acids in the binding through the free binding energy decomposition and declared the interactions variation of compound **10** inside VEGFR-2 via the Protein–Ligand Interaction Profiler (PLIP). Being new, its molecular structure was optimized by DFT. The DFT studies also confirmed the binding mode of compound **10** with the VEGFR-2. ADMET (in silico) profiling indicated the examined compound’s acceptable range of drug-likeness. The designed compound was synthesized through the condensation of *N*-(4-(hydrazinecarbonyl)phenyl)benzamide with *N*-(4-acetylphenyl)nicotinamide, where the carbonyl group has been replaced by an imine group. The in-vitro studies were consonant with the obtained in silico results as compound **10** prohibited VEGFR-2 with an IC_50_ value of 51 nM. Compound **10** also showed antiproliferative effects against MCF-7 and HCT 116 cancer cell lines with IC_50_ values of 8.25 and 6.48 μM, revealing magnificent selectivity indexes of 12.89 and 16.41, respectively.

## 1. Introduction

It is forecasted by the world health organization (WHO) that cancer is the second leading cause of death. The most widespread cancer in 2020 (on the basis of the incidence of the new cases) was breast cancer, reaching 2.26 million reported cases [1]. Against this background, the development of chemotherapeutic agents, which interact with specific molecular targets and subsequently damage cancer cells, is a challenging endeavor for medicinal chemists. The progression of tumors and their spreading are linked to the development of angiogenesis [2,3]. Therefore, anti-angiogenesis treatments are considered the most promising methods to defeat cancer [4]. The vascular endothelial growth factor (VEGF) pathway is an important regulator of the angiogenesis process and led to the discovery of a wide array of chemotherapeutic agents [5,6]. Vascular endothelial growth factor receptor-2 (VEGFR-2) is a critical transmembrane tyrosine kinase receptor in cancer management. VEGFR-2 orchestrates critical steps in cell proliferation, division, motility, adhesion, and angiogenesis [7,8], such that, reducing the VEGFR-2 signaling cascade reduces the proliferation of cancer cells [9]. Researchers have developed safe and selective drugs that specifically target the VEGFR-2 receptors in tumor cells without affecting normal tissues due to the overexpression of those receptors in cancer cells [10]. In the last decade, there have been many efforts conducted to discover new VEGFR-2 inhibitors [11,12,13]

Several types of software are used in the in-silico (computer-based) chemistry to integrate mathematical and theoretical principles to investigate and answer chemical problems. This approach is widely employed in the pharmaceutical industry to understand how potential drugs interact with various biomolecules [14,15,16].

Further, several experiments employed computational chemistry and have been conducted to explore molecular design [17], MD simulations [18], ADMET [19], toxicity [20], DFT [21], structural similarity [22], and pharmacophore assessment [23].

In this research, we incorporate our past experiences in computer-based chemistry and medicinal chemistry to develop an effective anticancer nicotinamide derivative attacking VEGFR-2. The nicotinamide lead compound was designed while keeping the general features of VEGFR-2 inhibitors. Then, its potential against VEGFR-2 was declared by several computational experiments (molecular docking, MD simulations, MM-GBSA, and DFT). Finally, several in-vitro experiments were carried out to judge the accomplished studies.

### Rationale

Sorafenib **I** [24], altiratinib **II**, regorafenib **III**, and SKLB-610 **IV** are reported VEGFR-2 inhibitors. The nicotinamide derivatives (compounds **V** and **VI**) were previously synthesized by our team and exhibited promising anti-proliferative VEGFR-2 inhibitory activities. In addition, these compounds exhibited an apoptotic effect. Compounds **I–VI** presented in Figure 1 have a pyridine moiety that occupies the hinge region at the active site of VEGFR-2.

These compounds possess four essential pharmacophoric features that form maximal fitting against the VEGFR-2 active site. These features include hetero-aromatic nucleus, linker moiety, hydrogen bond donor, hydrogen bond acceptor atoms (pharmacophore moiety), and terminal hydrophobic group. Respectively, these four features occupy the hing region, gatekeeper region, DFG motif region, and allosteric binding pocket of the ATP binding site of VEGFR-2 [25].

In the current work, compound **V** was used as a lead compound to do chemical modification at three sites. The first site is the phenyl ring of the linker moiety. The lead compound has a para disubstituted phenyl ring as the pharmacophore moiety is attached to the position 4 of the phenyl ring. In the modified compound, the pharmacophore moieties are attached to position 3 of the phenyl ring to become a meta-substituted derivative. This modification will change the orientation of the new compound and may facilitate its efficient binding in the active site. The second site is the pharmacophore moiety as we replaced the hydrazone moiety (in the lead compound) with the formylhydrazone one (in the modified compound). This modification may increase hydrogen bonding interaction at the DFG motif region. The third site at which a modification took place is the terminal hydrophobic tail. We applied the extension strategy (addition of extra function group). In this strategy, a benzamide moiety was substituted at position 4 of the terminal phenyl ring. This modification may increase the chance to form extra hydrophilic and hydrophobic interactions in the allosteric binding site (Figure 2).

## 2. Results

### 2.1. Molecular Docking Simulations against VEGFR-2

To investigate the possible binding pattern of compound **10** in the active site of VEGFR-2, molecular docking simulations were performed using MOE software. The docking calculations used the high-resolution crystal structure of VEGFR-2 (PDB ID: 2OH4) as a target enzyme. First, validation of the docking procedure was succeeded as present in Figure 3 by the re-docking of the co-crystallized ligand and production of a low RMSD value (0.68 Å).

The obtained results of the docking process indicated that the outlined binding mode of sorafenib was consistent with what is reported in the literature [26] (Figure 4).

The best docking pose of compound **10** was nominated as the most probable binding conformation. The docked compound showed promising binding patterns in the VEGFR-2 binding site, interacting with the key amino acids. The pyridine ring is accommodated in the hinge region, achieving hydrogen bonding interaction with Cys917 besides three 2 π interactions with Leu838, Leu1033, and Ala864. The 1-phenylethan-1-imine moiety represented a linker moiety that occupied the linker region to form eight hydrophobic interactions with Ala864, Val846, Val912, Lys866, Leu887, and Val914. As well, the amide moiety of the hydrazone arm formed a hydrogen bond with the carboxylate moiety of Glu883 (1.84 Å) and another hydrogen bond with the NH of Asp1044 (2.11 Å) in the DFG region. The terminal *N*-phenylbenzamide interacts with the hydrophobic side chains of the amino acids (Leu887 and Ile886) lining the allosteric pocket (Figure 5).

### 2.2. MD Simulations Results

We performed molecular dynamics simulations for compound 10 to study the strength of binding to the VEGFR-2 protein. The trajectory was used to extract the RMSD (Figure 6A), RMSF (Figure 6B), SASA (Figure 6C), RoG (Figure 6D), the change in the number of hydrogen bonds (Figure 6E), and the distance between the center of masses between compound 10 and VEGFR-2 protein (Figure 6F). The RMSD of the compound 10-VEGFR-2 complex can be divided into two regions. The first one contains approximately the first 42 ns with an average of 2.19 Å before increasing in the rest of the simulation to values with an average of 3.85 Å. On the other hand, the RMSD of compound 10 shows a relatively large fluctuation in the first 44 ns before coming to stable values around 2.8 Å. The reason for the increase in the RMSD after 42 ns is the large motion of the K1053:L1065 loop as shown in the RMSF values (Figure 6B). In addition, the terminals show very large fluctuations reaching 12 Å. On the other hand, most of the amino acids have fluctuations less than 2 Å. The values of SASA (average = 17765 Å^2^), RoG (average = 20.7 Å), and the change in the number of hydrogen bonds (average = 68 bonds) show that the compound 10-VEGFR-2 complex conformation is stable with no unfolding or folding occurring. The distance between the center of mass of the ligand and the VEGFR-2 protein indicates that the ligand is bound to the protein during the simulation with an average of 8.36 Å.

### 2.3. MMGBSA

To measure the strength of binding, the gmx_MMPBSA library was utilized. Figure 7 shows the values of energy components of the MMGBSA and their standard deviations. The binding in the compound 10-VEGFR-2 complex is mostly due to the Van Der Waals interaction (average of −57.11 Kcal/Mol) followed by the electrostatic interactions (average of −23.83 Kcal/Mol) and total binding energy of −38.36 Kcal/Mol. Amino acid contribution to the binding was measured by the decomposition of the MMGBSA to know which amino acids are contributing most to the interaction (Figure 8). Nine amino acids show a contribution to the binding, with values less than −1 Kcal/Mol. V846, I886, L887, V914, C1022, I1023, C1043, D1044, and F1045 show binding contributions of −1.32 Kcal/Mol, −1.27 Kcal/Mol, −1.31 Kcal/Mol, −1.03 Kcal/Mol, −2.09 Kcal/Mol, −2.11 Kcal/Mol, −3.81 Kcal/Mol, −1.87 Kcal/Mol, and −1.04 Kcal/Mol, respectively.

To know the number and types of interaction, the trajectory was clustered and for each cluster, a representative frame was obtained that was used with the PLIP webserver. Table 1 shows the number and types of interactions for each frame. The predominant interaction is the hydrophobic interaction in all the representative frames, which supports the value of the Van Der Waals component in MMGBSA analysis. In addition, PLIP outputs the .pse file that shows the 3D interaction pattern for each representative frame (Figure 9).

### 2.4. Density Function Theory (DFT)

#### 2.4.1. Structure Optimization

The DFT/B3LYP approach was used in the current work to perform quantum chemical computations to optimize the structures of the title compound. The DFT (B3LYP) method with 6-311G++(d,p) basis set was applied in this test [27]. The chemical structure of the selected compound is formed by the condensation reaction of *N*-(4-(hydrazinecarbonyl)phenyl)benzamide with *N*-(4-acetylphenyl)nicotinamide, where the carbonyl group was replaced by the azomethine or imine group. The optimized structure is numbered and represented in Figure 10. The DFT calculations revealed that the azomethine bond length (C16-N18) was found to be 1.28959Å, whereas the two angles located on the sides of the azomethine bond, (C17C16N18) and (C16N18N19), were found to be 123.11240° and 118.90226°, respectively.

#### 2.4.2. Frontier Molecular Orbital Analysis and Global Chemical Descriptors

The molecular orbital energies are an effective tool in quantum chemistry for explaining a molecule’s electric and optical characteristics. The frontier molecular orbitals (FMO) are two significant orbitals called HOMO and LUMO. Both orbitals located at the outermost borders of the molecules’ electrons are used to define conjugated molecules. Specifically, the excitation of an electron from the highest occupied molecular orbital (HOMO) to the lowest unoccupied molecular orbital (LUMO) is explained by the FMO. The electron-donor character is measured by the HOMO energy (E_HOMO_), whereas the electron-acceptor character is measured by the LUMO energy (E_LUMO_). Greater HOMO energy value and lower LUMO energy value correspond to higher electron-donor capacity and lower electron-acceptance resistance, respectively.

The frontier molecular orbitals are depicted in Figure 11, where the negative and positive phases of the molecule are shown by the color codes of green to red. In Figure 11, HOMO is localized on most atoms of the molecule except for some hydrogen atoms, terminal phenyl, and pyridine rings, while LUMO is delocalized over the entire title molecule except for a few carbon and hydrogen atoms of the molecule.

The molecule stability concerning subsequent chemical reactions is suggested by the FMO energy gap (E_gap_), which is important. The E_gap_ of HOMO and LUMO as well as other electronic characteristics of the molecule under consideration were computed and tabulated in Table 2. The global chemical reactivity descriptors of molecules such as ionization potential (IP), chemical potential (µ), maximal charge acceptance (∆N_max_), global chemical hardness (η), global chemical softness (σ), energy change (∆E), electrophilicity (ω), electronegativity (χ), and electron affinity (EA) were estimated according to Koopmans’ theory [28].
IP = –E_HOMO_
EA = –E_LUMO_
µ = (IP + EA)/2
H = (IP − EA)
Χ = −η
Ω = µ^2^/(2 η)
Σ = 1/η
∆N = −(μ/η)
∆E = −ω
E_gap_= E_LUMO_–E_HOMO_

As shown in Table 2, all computed reactivity parameters of the title compound are listed and the energetic characteristics, including the dipole moment (Dm) and optimization energy (TE), were also determined and listed. From Figure 11 and Table 2, the E_gap_ between HOMO and LUMO is calculated to be 4.415 eV, which is relatively small [29]. A compound with a small frontier orbital gap means a high polarizable and chemically reactive compound. Such finding explains the inhibition ability of compound 10 against VEGFR-2.

The concept of a molecule’s chemical reactivity is closely related to theoretical chemistry, which is established on the FMO theory. The global reactivity descriptor, which indicates knowledge about the molecules’ reactivity or behavior, may be obtained using the DFT with amazing success. The calculated chemical descriptors in Table 2 showed that compound **10** is chemically reactive to act as a good inhibitor toward VEGFR-2 [30].

#### 2.4.3. The Total Density of State (DOS) and Electron Density Maps

Because of the possibility of quasi-degenerate energy levels at the border area, consideration of only LUMO and HOMO may not produce a meaningful characterization of frontier orbitals. For this reason, the python program GaussSum3.0 was used to depict the total density of state (TDOS). Convoluting the molecular orbital data generated by Gaussian led to the calculation and generation of the density of states or DOS. It offers a visual depiction of the compositions of molecular orbitals and how they affect chemical bonding. Figure 12 displays the estimated and obtained TDOS pictogram.

The molecular electrostatic potential (MEP) is a crucial means for examining and correlating the link between a molecule’s physicochemical properties and molecular chemical structure, including drugs and biomolecules. It is well known that every chemical system generates an electrostatic potential surrounding it. The electron density was mapped across the electrostatic potential surface of the selected molecule, which shows the molecular shape, size, and charge distribution. The electron density provides information about compound **10**’s interactions with one another while electrostatic potential is widely utilized for estimating reactivity and hydrogen bonding as well as inter and intra-molecular interaction forces of chemical systems. The MEP gives a visual method to show the relative polarity of the molecule and the ESP values of a molecule illustrated by various colors: red, green, and blue, which represent zones of the most negative, zero, and most positive electrostatic potential, respectively. Blue regions denote the strongest attraction, red zones point out the strongest repulsion, and green areas indicate neutral electrostatic potential. To determine reactive reaction sites of nucleophilic and electrophilic attack for the title compound, both the total electron density (TED) and electrostatic potential (ESP) are illustrated in Figure 13a,b, respectively. In compound **10**, the three carbonyl groups are the most electronegative chemical sites that facilitate the electrophilic attack of amino acids. In addition, the blue areas at N-H groups indicated the most positive sites, where donor atoms of amino acids donate these sites with electrons through the nucleophilic attack. The yellow regions at the N-pyridine atom denote atoms with moderate electronegativity. Furthermore, the direction of inhibition ability of the title molecule on the electrophilic amino acids is illustrated by the electrostatic surface potential (ESP) (Figure 13b), and it is the same orientation as the *N*-pyridine atom and the three carbonyl groups.

### 2.5. ADMET Profiling Study

Compound **10**’s ADMET parameters were computed applying Discovery studio 4.0. software using sorafenib as the reference molecule. Compound 10 and sorafenib showed striking similarities in ADMET results (Figure 14) as both compounds were found to have a very low ability to pass the BBB. The CNS toxicity may be diminished with the systemic action of this compound. Also, it showed low aqueous solubility and good intestinal absorption levels. To enhance its aqueous solubility, chemical modification or nano-formulation are recommended. The good absorption rate may increase the distribution of this compound and consequently its biological effect. Interestingly, both compounds were foreseen as non-inhibitors of the cytochrome P-450, CYP2D6. The hepatotoxicity may be absent upon administration of this compound. Finally, this compound was expected to bind against plasma protein by more than 90%.

### 2.6. In Silico Toxicity Studies

Five parameters of toxicity were estimated computationally in accordance with the toxicity models built in the Discovery studio software. The employed models are: carcinogen potential TD_50_ in a rat model (TD_50_-M); Ames Mutagenicity (Am-M), which predict if the tested compound is mutagenic or not; DTP prediction (DTP); rat maximum fed tolerated dose (MFTD-R); oral LD_50_ in rats (R-O- LD_50_); chronic LOAEL in rats (LOAEL-R); and skin and eye irritation potential. As demonstrated in Table 3, the designed compound was predicted to be much safer than sorafenib.

### 2.7. Chemistry

Our design indicated that compound **10** is highly able to bind with and inhibit VEGFR-2. Then, the conducted computational experiments confirmed the proposed binding ability. Consequently, compound **10** was synthesized to examine experimentally the design and the in-silico outputs.

The reactions illustrated in Figure 1 outlined the synthesis of the target pyridine-based derivative **10**. Initially, acylation of nicotinic acid **2** using thionyl chloride gave nicotinoyl chloride **3** [26]. Nicotinoyl chloride **3** was then reacted with 4-aminoacetophenone to afford the corresponding nicotinamide derivative **4**. On the other hand, esterification of 4-aminobenzoic acid **5** was achieved simply via refluxing in a methanol/sulfuric acid mixture to produce the corresponding ester **6** [31]. The produced methyl 4-benzamidobenzoate, **6**, underwent a benzoylation reaction by the drop-wise addition of benzoyl chloride **7** in a DCM/TEA mixture at 0 °C to afford the corresponding benzoyl derivative **8** according to the reported methods [32]. Refluxing of compound **8** with hydrazine hydrate in absolute ethanol produced the corresponding acid hydrazide derivative **9** [33]. Finally, compound **9** was then condensed with compound **4** to give the final target candidate **10**.

Spectral analyses for the synthesized compound **10** confirmed its structure. The ^1^H NMR revealed the presence of a characteristic singlet signal at 2.42 ppm corresponding to the CH_3_ group. Furthermore, ^1^H NMR exhibited the presence of three downfield singlet signals at 11.50, 10.76, and 10.66 ppm corresponding to the three amidic protons. Finally, the ^13^C NMR spectrum was also consistent with the assigned structure of the synthesized compound.

### 2.8. Biological Results

#### 2.8.1. VEGFR-2 Prohibition

To confirm the mechanism of action of compound **10** as a VEGFR-2 inhibitor, it was tested for its in-vitro inhibitory potential. The results of VEGFR-2 inhibition revealed that compound **10** can inhibit VEGFR-2 with a potency that was comparable to that of sorafenib. Compound **10** exhibited an IC_50_ value of 51 nM, while sorafenib’s IC_50_ value was 35 nM. The in-vitro results corresponded with the results of the in silico studies, confirming the great potential of compound **10** for prohibiting VEGFR-2. From different in-silico studies (docking and MD simulations) and in-vitro VEGFR-2 inhibition, we can conclude that the synthesized compound has good efficiency to inhibit the kinase activity of VEGFR-2. To validate this proposed mechanism of action, in-vitro cytotoxicity against two tumor cell lines was conducted.

#### 2.8.2. Cytotoxicity

To assess the applicability of compound **10**’s ability to inhibit VEGFR-2, in-vitro cytotoxicity tests were performed using MCF-7 and HCT 116 as cancer models. Sorafenib was used as the reference drug.

Figure 15 and Table 4 show that compound **10**’s potentiality to prevent the growth of MCF-7 cancer cell lines was very near (IC_50_ = 8.25 μM) to that of sorafenib. Interestingly, compound 10 inhibited the HCT 116 cancer cells with a stronger activity (IC_50_ = 6.48 μM) than sorafenib (IC_50_ = 7.28 μM).

#### 2.8.3. Safety Assessment

In order to verify the obtained safety results of the conducted computational models, the cytotoxic activity of compound **10** was determined against normal human cell lines (W138) to demonstrate the safety and determine its selectivity against cancer cell lines. Compound **10** demonstrated a very high level of safety against normal human W138 cells with an IC_50_ value of 106.38. These results indicate the excellent selectivity indexes of compound **10** against MCF-7 and HCT 116 cancer cell lines of 12.89 and 16.41, respectively.

## 3. Experimental

### 3.1. Docking Studies

The molecular docking was conducted for compound 10 against VEGFR-2 [34,35] by MOE2014 software (Montreal, QC, Canada). Appendix A provide a thorough explanation.

### 3.2. MD Simulations

CHARMM-GUI web server (Bethlehem, PA, USA) was employed and GROMACS 2021(Uppsala, Sweden) was used as an MD engine. Appendix A provide a thorough explanation.

### 3.3. MM-GBSA

The Gmx_MMPBSA package(Uppsala, Sweden) was used. Appendix A provide a thorough explanation.

### 3.4. DFT

Gaussian 09 and GaussSum3.0 programs (Gaussian, Inc., Wallingford CT, USA, 2009) were used. Appendix A provide a thorough explanation.

### 3.5. ADMET Studies

ADMET profile was determined by Discovery Studio 2016 (Vélizy-Villacoublay, France) [36]. Appendix A provide a thorough explanation.

### 3.6. Toxicity Studies

The toxicity profile was determined by Discovery Studio 2016 (Vélizy-Villacoublay, France) [37]. Appendix A provide a thorough explanation.

### 3.7. General Procedure for the Synthesis of Compound **10**

In a round bottom flask containing absolute ethanol (25 mL), *N*-(3-acetylphenyl) nicotinamide **4** (0.25 g, 0.001 mol) and *N*-(4-(hydrazinecarbonyl) phenyl)benzamide (0.26 g, 0.001 mol) were mixed and well dissolved. The whole mixture was then refluxed for 6 h using catalytic drops of glacial acetic acid. The reaction was monitored using TLC. After reaction completion, the mixture was concentrated and cooled. The collected product was filtered and purified by crystallization from methanol.


**
*(E)-N-(4-(1-(2-(3-Benzamidobenzoyl)hydrazono)ethyl)phenyl)nicotinamide*
**




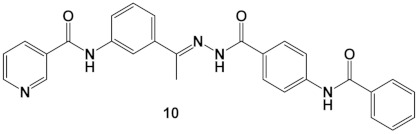



Off-white crystal (yield, 81%); m. p. = 205–207 °C; IR (KBr) *ν* cm^−1^: 3136 (NH), 3050 (CH aromatic), 2958, 2903 (CH aliphatic), 1677 (C=O); ^1^H NMR (400 MHz, DMSO-*d*_6_) δ 11.50 (s, 1H), 10.76 (s, 1H), 10.66 (s, 1H), 10.60 (s, 1H), 9.19 (s, 1H), 8.79 (d, *J* = 4.8 Hz, 1H), 8.40 (d, *J* = 8.1 Hz, 1H), 8.29 (s, 1H), 8.03–7.98 (m, 7H), 7.65–7.54 (m, 4H), 7.45 (dd, *J* = 8.1 Hz, 1H), 2.42 (s, 3H). ^13^C NMR (101 MHz, DMSO-*d*_6_) δ 166.5(C=O), 166.4(C=O), 165.9(C=O), 152.4(C2-pyridine), 149.1(Ph-C=N), 143.7(Ar-C), 139.4(Ar-C), 139.2(Ar-C), 136.3(Ar-C), 135.1(Ar-C), 135.0(Ar-C), 132.4(Ar-C), 129.1(Ar-C), 128.9(Ar-C), 128.3(Ar-C), 128.3(Ar-C), 125.8(Ar-C), 124.1(Ar-C), 120.1(Ar-C), 119.9(Ar-C), 118.9(Ar-C), 39.3(CH_3_); For C_28_H_23_N_5_O_3_ (477.52).

### 3.8. Biological Studies

#### 3.8.1. In Vitro VEGFR-2 Inhibition

In-vitro VEGFR-2 inhibition was performed using Human VEGFR-2 ELISA kit. Appendix A provide a thorough explanation.

#### 3.8.2. In Vitro Antiproliferative Activity

MTT procedure was employed. Appendix A provide a thorough explanation.

#### 3.8.3. Safety Assay

The non-cancerous cell lines, W138, were used. Appendix A provide a thorough explanation.

## 4. Conclusions

A new nicotinamide-based derivative was designed with the basic features of anti-VEGFR-2 drugs. The proposed compound is a modified analog of our previously discovered active candidate through substitution pattern variation, chain extension, and addition of an extra function group. The potential of binding was suggested for the designed derivative by molecular docking. Following, the accuracy and correctness of binding with the VEGFR-2 were verified and confirmed by MD simulations, MM-GBSA, and DFT. The MD simulations and MM-GBSA studies revealed that the proposed compound has great stability in the active site of VEGFR-2. The calculated DFT descriptors showef that the compound is chemically reactive to act as a good inhibitor toward VEGFR-2. Additionally, ADMET studies indicated the drug-likeness. In consequence, the designed nicotinamide compound was synthesized and indicated excellent in-vitro VEGFR-2 prohibitory potential (51 nM), promising cytotoxicity against MCF-7 and HCT 116 cancer cell lines with IC_50_ values of 8.25 and 6.48 μM demonstrating selectivity indexes of 12.89 and 16.41, respectively. The synthesized compound is considered a promising lead compound for further modification. In addition, deep biological investigations are recommended to reach a good insight into its activity at a molecular level.

## Data Availability

Data are available with corresponding authors upon request.

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
