# Peer review of "Design, Synthesis, Docking, DFT, MD Simulation Studies of a New Nicotinamide-Based Derivative: In Vitro Anticancer and VEGFR-2 Inhibitory Effects"

_molecules, 2022, doi:10.3390/molecules27144606_

Round 1

Reviewer 1 Report

Reference are not exactly wriiten according to journal requirements

Information in Materials and Methods (Experimental) can be supplemented. Supplementary data is OK but some can be introduced in the text of the manuscript

Future perspective must be discussed. Comclussion is superficially written

There are spelling and syntax errors in the text

Mechanism of action of compounds should be discussed

Author Response

The authors would like to thank the reviewer for his valuable comments that made the manuscript much better

  • References are not exactly written according to journal requirements.

Response: Modified

  • Information in Materials and Methods (Experimental) can be supplemented. Supplementary data is OK, but some can be introduced in the text of the manuscript.

Response: We agree with the reviewer. However, we have to do that to keep the plagiarism level less than 25% according to ithunticate. We believe that plagiarism is a serious concern to the authors as well as the journal.

  • Future perspective must be discussed. Conclusion is superficially written.

Response: DONE

  • There are spelling and syntax errors in the text

Response: The manuscript was revised thoroughly

  • Mechanism of action of compounds should be discussed.

Response: DONE

Reviewer 2 Report

The manuscript molecules-1822310 devoted the actual field of organic and medicinal chemistry, namely the design of new VEGFR-2 inhibitors and can be interested to the specialists working in this field. The authors’ opinion is clear and based on a good experimental material. I am personally impressed by the structure of the article, the systematization of scientific data and the sequence of its presentation. However, it needs minor revision before publication.

To improve the quality and perception of the manuscript I would suggest paying attention to following comments:

1.      Data of 13C NMR (experimental part) should be to 1 dp but not to 2 dp.

2.      The assignment of the 13C NMR data would be an added value.

3.      References list should be carefully checked and journal style policy should be strictly followed (journal abbreviations, DOI, etc).

4.      There are some grammar and orthographical errors in the manuscript, which should be corrected

My decision is minor revision.

Author Response

The authors would like to thank the reviewer for his valuable comments that made the manuscript much better

  • Data of 13C NMR (experimental part) should be to 1 dp but not to 2 dp.

Response: DONE

  • The assignment of the 13C NMR data would be an added value.

Response: DONE

  • References list should be carefully checked, and journal style policy should be strictly followed (journal abbreviations, DOI, etc).

Response: DONE

  • There are some grammar and orthographical errors in the manuscript, which should be corrected.

Response: The manuscript was revised thoroughly

Reviewer 3 Report

Journal Name: Molecules

Title:  Design, synthesis, docking, DFT, MD simulation studies of a new nicotinamide derivative: in vitro anticancer and VEGFR-2 inhibitory effects.

The authors have made detailed research on “theoretical analysis of new nicotinamide derivative for in vitro anticancer and VEGFR-2 inhibitory effects”. This is interesting theoretical research and it is suitable for MDPI Molecules. I recommend its publication with minor revision. Here I am mentioning my detailed comments.

1.      I suggest authors cite relevant articles on MDPI Molecules so that it will prove the article’s suitability to publish on MDPI molecules.

2.      In the title author can mention the commonly used name of the nicotinamide derivative. Just saying the new nicotinamide derivative does not look good.

3.      The authors need to explain how they are confirming (cross-checking) the results of MD results in detail.

4.      The resolution of Fig. 8 has to be improved.

5.      The authors need to show the respective frequencies of the selected Schiff base compound to confirm the optimization of the molecule.

6.      The authors need to mention the utilized basis set in section 2.4.1

7.      The authors need to explain the importance of ADMET profiling study

8.       The authors need to give the respective equations and references for the below-mentioned global electron transfer properties maximal charge acceptance (∆Nmax), global chemical hardness (η), global chemical softness (σ), energy change (∆E), electrophilicity (ω), electronegativity (χ), and electron affinity.   

9.      Graphical abstract is not visible

  1. Please check the grammatical and syntax error

Author Response

The authors would like to thank the reviewer for his valuable comments that made the manuscript much better

  1. I suggest authors cite relevant articles on MDPI Molecules so that it will prove the article’s suitability to publish on MDPI molecules.

Response: DONE

  • In the title author can mention the commonly used name of the nicotinamide derivative. Just saying the new nicotinamide derivative does not look good.

Response: DONE

  • The authors need to explain how they are confirming (cross-checking) the results of MD results in detail.

Response: We did not performed cross-checking. However, if the reviewer think it is necessary, we can do it in 7 working days. Please, let us know.

  • The resolution of Fig. 8 has to be improved.

Response: DONE

  • The authors need to show the respective frequencies of the selected Schiff base compound to confirm the optimization of the molecule.

Response: Thank you for pointing this important point. The DFT calculations were performed with the GAUSSIAN 09 package, using the B3LYP exchange correlation function using the B3LYP/6-311++G(d,p) basis set without any symmetry constraint. The frequency calculations were performed at the same level to make sure the global minima by without imaginary frequency and the lack of imaginary frequencies provided evidence for full optimization of the structure. After confirming the absence of imaginary frequency in the generated structures, the resultant optimal conformations were made as input files for the further spectroscopic and other electron density computations following the same calculation scheme as described in the optimization.

  • The authors need to mention the utilized basis set in section 2.4.1.

Response: DONE                

  • The authors need to explain the importance of ADMET profiling study.

Response: DONE

  • The authors need to give the respective equations and references for the below-mentioned global electron transfer properties maximal charge acceptance (∆Nmax), global chemical hardness (η), global chemical softness (σ), energy change (∆E), electrophilicity (ω), electronegativity (χ), and electron affinity.   

Response:  DONE

  • Graphical abstract is not visible.

Response: Graphical abstract was provided

  • Please check the grammatical and syntax error.

Response:  Done

Reviewer 4 Report

Manuscript presented by Eslam B. Elkaeed et al. shows a comprehensive study of nicotinamide-based derivative compound with the pharmacological properties. In the article Authors focus on synthesis and in vitro study as well as quantum-chemical calculation based on DFT. An already well written and prepared manuscript. Easy to read and follow. I recommend the article to publish but first the paper should be improve. My decision – reconsider after minor revision. Comments to be considered, in order to further improve the manuscript quality:

(1)   Please delete not necessary words like “captivatingly” ect in all document. It is scientific text form, not informal and these kind of terms add nothing to the main discussion.

(2)   Abstract should have information about name of compound and its synthesis method. Add it.

(3)   The conclusion need to be re-written, are too general. The flow of this chapter is poor. 

(4)   Reorganise the "experimental" section.

(5)   Please use the template Molecules. Correct it. What is more the reference style does not match this journal. It should be changed.

(6)   The English correction is necessary.

Author Response

The authors would like to thank the reviewer for his valuable comments that made the manuscript much better

  • Please delete not necessary words like “captivatingly” ect in all document. It is scientific text form, not informal and these kinds of terms add nothing to the main discussion.

Response: Done

  • Abstract should have information about name of compound and its synthesis method. Add it.

Response: Done

  • The conclusion needs to be re-written, are too general. The flow of this chapter is poor. 

Response: Done

  • Reorganize the "experimental" section.

Response: Done

  • Please use the template Molecules. Correct it. What is more the reference style does not match this journal. It should be changed.

Response:  Done

  • The English correction is necessary.

Response:  The manuscript was revised thoroughly